# Association between serum periostin levels and the severity of arsenic-induced skin lesions

Moriom Khatun[1], Abu Eabrahim Siddique[2], Abdus S. Wahed[3], Nazmul Haque[1], Selim Reza Tony[1], Jahidul Islam[1], Shahnur Alam[4], Md. Khalequzzaman Sarker[5], Isabela Kabir[6], Shakhawoat Hossain[1], Daigo Sumi[7], Zahangir Alam Saud[1], Aaron Barchowsky[4], Seiichiro Himeno[7,8], Khaled Hossain[1]*

1 Department of Biochemistry and Molecular Biology, University of Rajshahi, Rajshahi, Bangladesh, 2 Department of Biological Sciences, State University of New York at Buffalo, Buffalo, New York, United States of America, 3 Department of Biostatistics, University of Pittsburgh, Pittsburgh, Pennsylvania, United States of America, 4 Department of Environmental and Occupational Health, University of Pittsburgh, Pittsburgh, Pennsylvania, United States of America, 5 Institute of Biological Sciences, University of Rajshahi, Rajshahi, Bangladesh, 6 Labaid Specialized Hospital, Dhaka, Bangladesh, 7 Laboratory of Molecular Toxicology, Faculty of Pharmaceutical Sciences, Tokushima Bunri University, Tokushima, Japan, 8 Division of Health Chemistry, School of Pharmacy, Showa University, Tokyo, Japan

☯ These authors contributed equally to this work.
* khossain@ru.ac.bd

**Data Availability Statement:** All relevant data are within the paper, Supporting Information files, and at Figshare (doi.org/10.6084/m9.figshare.21671057).

## Abstract

Arsenic is a potent environmental toxicant and human carcinogen. Skin lesions are the most common manifestations of chronic exposure to arsenic. Advanced-stage skin lesions, particularly hyperkeratosis have been recognized as precancerous diseases. However, the underlying mechanism of arsenic-induced skin lesions remains unknown. Periostin, a matricellular protein, is implicated in the pathogenesis of many forms of skin lesions. The objective of this study was to examine whether periostin is associated with arsenic-induced skin lesions. A total of 442 individuals from low- (n = 123) and high-arsenic exposure areas (n = 319) in rural Bangladesh were evaluated for the presence of arsenic-induced skin lesions (Yes/No). Participants with skin lesions were further categorized into two groups: early-stage skin lesions (melanosis and keratosis) and advanced-stage skin lesions (hyperkeratosis). Drinking water, hair, and nail arsenic concentrations were considered as the participants' exposure levels. The higher levels of arsenic and serum periostin were significantly associated with skin lesions. Causal mediation analysis revealed the significant effect of arsenic on skin lesions through the mediator, periostin, suggesting that periostin contributes to the development of skin lesions. When skin lesion was used as a three-category outcome (none, early-stage, and advanced-stage skin lesions), higher serum periostin levels were significantly associated with both early-stage and advanced-stage skin lesions. Median (IQR) periostin levels were progressively increased with the increasing severity of skin lesions. Furthermore, there were general trends in increasing serum type 2 cytokines (IL-4, IL-5, IL-13, and eotaxin) and immunoglobulin E (IgE) levels with the progression of the disease. The median (IQR) of IL-4, IL-5, IL-13, eotaxin, and IgE levels were significantly higher in the early-and advanced-stage skin lesions compared to the group of participants without

**Funding:** Dr. Khaled Hossain: Two grants from Ministry of Science and Technology, Government of the People's Republic of Bangladesh [grant nos. 39.009.006.01.00.042.2012-2013/ES-21/558 and 2007-2008/ BS-135/176/1(5)] and a Rajshahi University Grant [grant no. A-795/6-109 (Research)]. Seiichiro Himeno: Two Japan Society for the Promotion of Science (JSPS) KAKENHI grants [grant nos. 16H05834 and 24310048]. Aaron Barchowsky: A grant from the United States National Institute of Environmental Health Sciences [grant no. R01ES033519]. The funders had no role in study design, data collection and analysis, decision to publish, or preparation of the manuscript.

**Competing interests:** The authors have declared that no competing interests exist.

skin lesions. The results of this study suggest that periostin is implicated in the pathogenesis and progression of arsenic-induced skin lesions through the dysregulation of type 2 immune response.

## Introduction

Arsenic is a potent environmental toxicant and human carcinogen. Because of its strong carcinogenicity, International Agency for Research on Cancer (IARC) has ranked arsenic as a class I human carcinogen. Arsenic is ubiquitously present in the environment and poses a serious threat to public health in many countries. However, there may be no greater incidence of arsenic-promoted diseases than in Bangladesh, where millions of people are exposed to arsenic, mainly via drinking water, at concentrations greater than the permissive limit (10 μg/L) set by the World Health Organization (WHO).

Prolonged exposure to moderate to high concentrations of arsenic causes skin lesions. Arsenic-induced skin lesions are mainly manifested as melanosis and keratosis. Skin lesions are the most common adverse effects and hallmark features of prolonged human exposure to arsenic. Arsenic-induced cutaneous toxicity first appears as melanosis (i.e., hyperpigmentation of the skin) [1–4]. Melanosis manifests as diffuse and dark brown or blackish lesions or spotted dense pigmentation on the different parts of the body, including the chest, hands, legs, shoulder, and back [5]. Melanosis is followed by hyperkeratosis on the palms, soles, and dorsal sides of the hands and legs if exposure to arsenic continues [1, 6, 7]. Hyperkeratotic skin lesions can transform into a variety of cancers, such as basal cell carcinoma, Bowen's disease, and squamous cell carcinoma in the course of time [8–11]. Because of this phenomenon, skin lesions are often considered precursors of skin cancer [11, 12]. It has been reported that patients with arsenic-induced skin lesions are more prone to develop lung cancer, indicating the role of skin lesions in predicting subsequent other forms of internal malignancies [13–15]. Skin lesions, especially hyperkeratoses decrease the overall quality of life of patients with intense pain in the area of lesions, discomfort, anxiety, depression, and many other mental burdens [16, 17]. Hyperkeratotic skin lesions on soles also decrease mobility as the patients feel pain during walking. Because of their dire consequences, skin lesions may be more disastrous than any other diseases caused by arsenic, based on disability adjusted life years. Therefore, understanding the underlying mechanism and prevention of skin lesions is an important issue from a public health perspective.

Periostin, a matricellular protein, is involved in many physiological processes including wound healing, embryonic development, and osteoblastic proliferation and differentiation [18–20]. In contrast to its roles in the normal physiological process, aberrant expression of periostin is found in the cancers in skin, lung, prostate, bladder, and pancreases, as well as melanoma and cutaneous T cell lymphoma [21–28]. Periostin changes the tumor microenvironments and plays multiple roles in tumor initiation, progression, and metastasis [29–32].

Periostin exhibits protective effects on skin injury by acting on keratinocytes and fibroblasts under normal physiological conditions [33, 34]. However, growing evidence suggests that overexpression of periostin is involved in the pathogenesis of many inflammatory skin diseases, including atopic dermatitis (AD), psoriatic lesion, and scleroderma [35–37]. Periostin promotes skin fibrosis and reflects the severity of skin fibrosis in patients with systemic sclerosis [34, 38]. As a downstream event of type 2 immune response, periostin triggers skin-tissue remodeling in AD [39, 40]. Hyperkeratotic skin lesions in AD have a distorted architecture of the skin epidermis with subcutaneous fibrosis that results from chronic inflammation and

dysregulated immune response [41–44]. Although periostin is involved in various kinds of skin pathologies, the roles of periostin in arsenic-induced skin lesions remain unknown. Therefore, we explored the association between arsenic-induced skin lesions and serum periostin levels among the participants who were recruited in our previous study [45] from the low- and high-arsenic exposure areas in Bangladesh.

## Materials and methods

### Ethics statement

This study has been approved by Ethics Committee of Institute of Biological Sciences, University of Rajshahi, Bangladesh (No. 104/320/IAMEBBC/IBSc and 661/320/IAMEBBC/IBSc). The human participants who took part in this study provided written consent.

### Recruitment of study participants

The participants of this study were selected from our previous study [45]. The details of the study areas and the participants' recruitment process were described in our previous studies [45–47]. Briefly, we selected participants from high arsenic-exposure (arsenic-endemic) villages in rural Bangladesh that included Dutpatila, Jajri, Vultie, and Kestopur in the Chuadanga district, Marua in the Jessore district, Khemirdia in the Kushtia district, and Kazirpara in the Rajshahi district. We selected a low-exposure village, named Chowkoli from a northern district (Nagoan) as described in our previous studies [45–47] with no history of arsenic exposure. The total number of subjects in our previous study [45] was 442. Of them, 319 were from the high-exposure areas and 123 from the low-exposure area. The participants were 18–60 years old and had lived in their areas for at least five years. Including the present one, we have conducted a couple of studies [45, 47] using serum samples of the participants who were recruited in our previous study [46]. The overall response rate of the previous study [46] was 92% (93% from arsenic-endemic areas, and 91% from non-endemic area). We obtained relevant information such as age, sex, height, weight, smoking habit, education, occupation, income, and marital status through a paper-based structural questionnaire, as we described previously [46].

### Identification of skin lesions

Melanosis and keratosis are the major dermal manifestations of chronic exposure to arsenic [48]. Melanosis is manifested by many signs: diffuse melanosis (darkening of the skin on the different parts of the body such as the chest, hands, legs, shoulder, back, limbs, or palms of the hands), spotted pigmentation commonly appears on the chest, back, or the limbs and white and black spots on body (leucomelanosis) [2, 4, 49]. Keratosis is manifested by the following signs: bilateral thickening of the skin of the soles and hands, nodular keratosis (small protrusion emerged on the skin) on the palms and soles and spotted keratosis on the dorsal side of the hands, feet, and legs [5, 50–52]. Mild-form or early-stage keratosis appears on the palms and soles with rough nodules (grit-like character) which are not visible but can be identified by palpation [2, 12]. Moderate and severe forms of keratosis (hyperkeratosis) appear as large nodular keratoses (corn-like protrusion) which are readily visible [2, 4, 50]. Arsenic-induced dermal toxicity progresses from melanosis to keratosis and hyperkeratosis [1, 6, 53]. A physician team carefully checked the different parts of each participant's body and identified skin lesions. The arsenical skin lesion case diagnosed by the physician was further confirmed by a dermatologist. We categorized skin lesions into two groups based on the severity of dermal manifestations: i) early-stage skin lesions included all forms of melanosis (Fig 1A and 1B) and mild forms of keratosis (non-visible keratosis but felt by palpation) on palms and soles, and ii)

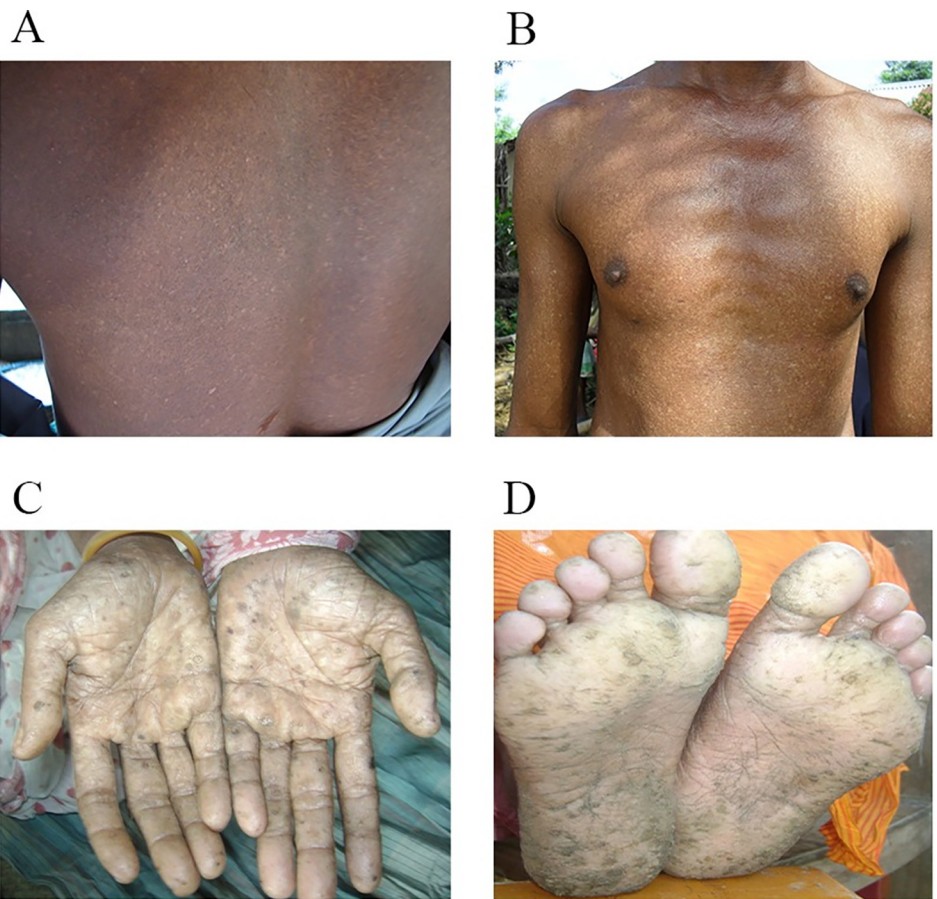

**Fig 1. Representative photographs of arsenic-induced skin lesions.** A. Leucomelanosis on the back, B. Diffuse pigmentation on the chest, C. Severe hyperkeratosis (large nodular keratosis) on the palms, and D. Severe hyperkeratosis on the soles.

advanced-stage skin lesions included moderate and severe forms of keratosis (visible hyperkeratosis) (Fig 1C and 1D). All the patients with severe forms of hyperkeratosis had characteristic features of melanosis. We did not find any subjects who had arsenical skin lesions in the low-exposure area.

## Exposure estimate

Arsenic concentrations in drinking water, hair, and nails of the participants were determined in our previous studies [45, 46] using inductively coupled plasma mass spectroscopy (ICP-MS) (Agilent 7700x, Japan) with the detection limit of $3.33 \times 10^{-3}$ ppb. The accuracy of the measurement was validated using appropriate certified reference materials. We used "river water" (National Metrology Institute of Japan [NMIJ] CRM 7202-a No. 347, National Institute of Advanced Industrial Science and Technology, Japan) as a certified reference material (CRM) for the validation of arsenic concentrations in the drinking water samples and used "human hair" (GBW09101, Shanghai Institute of Nuclear Research Academia Sinica, China) as a CRM for the measurement of arsenic in the biological samples (hair and nails). All CRMs and the samples were analyzed in triplicate and duplicate, respectively. The average value of triplicate "river water" was 1.161 ± 0.082 μg/L (reference value, 1.18 μg/L) and that of triplicate "human hair" was 0.278 ± 0.006 μg/g (reference value, 0.28 μg/g).

## Measurement of serum levels of periostin, type 2 cytokines and serum IgE

We used serum periostin, type 2 cytokines (IL-4, IL-5, IL-13, and eotaxin), and immunoglobulin E (IgE) data that were obtained in our previous studies [45, 47]. Serum periostin, IL-4, IL-5, IL-13, eotaxin, and IgE levels in our previous study were measured using enzyme-linked immunosorbent assay (ELISA) kits. Each sample was analyzed in duplicate, and the mean value was used for the final analysis. The members of our research team involved in the immunoassay were completely blind to the arsenic exposure levels until the final analysis.

## Statistical analysis

Participants with and without skin lesions were compared based on socio-demographic and other characteristics. Categorical variables (sex, smoking habit, education, and occupation) were summarized using frequency and percentages, and compared between the two groups using the Chi-square test. For continuous variables, first, the normality of the data was checked by a Q-Q plot, followed by a Kolmogorov–Smirnov test. For skewed continuous variables (income and BMI), median (25th percentile, and 75th percentile) was used as the summary and the Mann-Whitney U test was used to evaluate the difference in distribution between skin lesion and without skin lesion groups. Age, assessed to be normally distributed, was summarized using mean and standard deviations and compared between groups using an independent two-sample t-test. We also performed the Mann-Whitney U test to evaluate the difference in the distribution of arsenic exposure metrics (water, hair, and nail arsenic concentrations), and serum periostin levels between skin lesion and without skin lesion groups. The distribution of serum periostin levels across the three or four groups were graphically represented by boxplots and compared using the Kruskal-Wallis test followed by the Dunn-Bonferroni post hoc test. Age, smoking, and education status were considered as possible confounders since these variables were associated with the exposure and outcomes in the bivariate analysis at $p$-value $<0.1$. In addition, sex and BMI were also considered as confounders for skin lesions as reported by the previous studies [54, 55].

To assess the association of arsenic exposure and periostin levels with skin lesions, separate logistic regression models for each exposure variable (water, hair, and nail arsenic, and periostin) were fit with skin lesions (Yes/No) as the dependent variable and age, sex, BMI, smoking habit, and education as the adjusting factors. Results were reported using adjusted odds ratios, 95% confidence intervals, and $p$-values. Prior studies demonstrated a strong relationship between arsenic exposure and skin lesion [7, 51, 56, 57], and the above analyses showed a strong association between arsenic exposure and periostin levels, and in turn, between periostin levels and skin lesion. Thus, we conducted a causal mediation analysis to examine whether periostin levels mediate the relationship between arsenic levels and skin lesions. We represented the causal relationship using direct, and indirect (mediated through periostin) effects which were estimated using the standard Baron-Kenny regression-based approach [58, 59]. More specifically, the directed acyclic graph (DAG) presented in the S1 Fig was postulated. Consequently, a linear regression model was fit to model the association between the exposure (arsenic) and mediator (periostin) adjusting for other confounders (age, sex, BMI, smoking, and education), and logistic regression models were fit to model the association between outcome (skin lesions) and the exposure and the mediator. Three mediation analyses were conducted for the combination of three exposure markers (water, hair, and nail arsenic concentrations), and one outcome (skin lesions).

For the three-category ordinal skin lesions outcome, we first tested the homogeneity of the odds ratio (OR) in an ordinal logistic regression analysis. Since the homogeneity of the OR could not be established, we performed a multinomial logistic regression analysis to assess the

associations of arsenic exposure and serum periostin levels with the risk of skin lesions (early-stage and advanced-stage skin lesions) adjusting for the covariates (age, sex, BMI, smoking habit, and education). The results of these analyses were presented using adjusted ORs and 95% confidence intervals. For type 2 cytokines (IL-4, IL-5, IL-13, and eotaxin) and IgE, median (25th percentile, and 75th percentile) was used as the summary, and the Kruskal-Wallis test followed by the Dunn-Bonferroni post hoc test was performed to evaluate the difference in distribution among the groups of participants without skin lesions, early- and advanced-stage skin lesions.

All the statistical analyses for this study were conducted using SPSS (IBM Corp. Released 2015. IBM SPSS Statistics for Windows, Version 23.0. Armonk, NY: IBM Corp.) except for the causal mediation analysis, which was conducted using the SPSS PROCESS macro version 3.5 [60, 61]. In addition, the boxes and whisker plots with individual data points were created using GraphPad Prism 7.05 (GraphPad Software, La Jolla, CA).

## Results

### Descriptive characteristics of the groups of participants with and without skin lesions

Table 1 shows the participants' characteristics of the groups of participants with and without skin lesions. Out of 442, 223 (50.50%) participants had arsenic-induced skin lesions, while 219 (49.50%) had no skin lesions. Approximately ¼ (n = 112) of the total participants had advanced-stage diseases (visible hyperkeratosis along with the signs of melanosis). Participants with skin lesions were significantly older than those without skin lesions (mean 39.67 vs. 35.23 years, $p < 0.001$). The percentages of the educated participants and smokers in the skin lesions group were lower and higher, respectively, with borderline significant values compared to participants without skin lesions. A total of 91 (20.60%) participants were smokers and none of them were females. Sex, BMI, and other socio-demographic characteristics such as occupation, and monthly income did not differ significantly between the two groups. Most of the male participants in both groups were farmers, while most of the female participants were housewives.

### Comparisons of arsenic exposure levels between the groups of participants with and without skin lesions

Fig 2 shows the distributions of arsenic exposure levels (drinking water, hair, and nail arsenic concentrations) between the groups of participants with and without skin lesions. The median (IQR) of arsenic concentrations in drinking water, hair, and nails among the participants without skin lesions were significantly lower than those with skin lesions [Water As 3.20 (0.96, 82.73) vs. 167.94 (54.40, 263), $p < 0.001$; Hair As 0.56 (0.24, 2.03) vs. 3.10 (1.59, 6.32), $p < 0.001$; Nail As 1.83 (0.82, 3.78) vs. 7.26 (3.82, 14.10), $p < 0.001$] (Fig 2 and S1 Table).

### Arsenic exposure and serum periostin levels and their risk of skin lesions

Table 2 shows the associations of participants' arsenic exposure and periostin levels with the risk of skin lesions. Each log unit of water, hair, and nail arsenic concentrations were significantly associated with 3.41-, 10.22- and 15.66-fold increases in the odds of skin lesions (all $p < 0.001$), respectively, while each unit increase in periostin level was significantly associated with 1.03-fold increase ($p < 0.001$) in the odds of skin lesions. All the associations were adjusted for age, sex, BMI, smoking habit, and education level.

**Table 1. Descriptive characteristics of the groups of participants with and without skin lesions.**

| Parameters | All | Skin lesions | | p-value |
|---|---|---|---|---|
| | | No | Yes | |
| **Study participants [n, (%)]** | 442 | 219 (49.50) | 223 (50.50) | |
| **Sex [n, (%)]** | | | | |
| Male | 222 (50.20) | 111 (50.70) | 111 (49.80) | 0.849[†] |
| Female | 220 (49.80) | 108 (49.30) | 112 (50.20) | |
| **Age (years)[a]** | 37.47 ± 11.04 | 35.23 ± 11.08 | 39.67 ± 10.57 | <0.001[*] |
| **BMI (kg/m²)[b]** | 20.74 (18.70, 23.40) | 20.80 (18.92, 23.80) | 20.69 (18.50, 23.11) | 0.314[‡] |
| **Occupation [n, (%)]** | | | | |
| Farmers | 179 (40.50) | 88 (40.20) | 91 (40.80) | 0.779[†] |
| Housewives | 212 (48) | 103 (47) | 109 (48.90) | |
| Students | 17 (3.80) | 15 (6.80) | 2 (0.90) | |
| Business | 6 (1.40) | 1 (0.50) | 5 (2.20) | |
| Workers | 18 (4.10) | 8 (3.70) | 10 (4.50) | |
| +Others | 10 (2.30) | 4 (1.80) | 6 (2.70) | |
| **Education [n, (%)]** | | | | |
| No formal education | 246 (55.70) | 113 (51.60) | 133 (59.60) | 0.062 [†] |
| Primary (Below Grade 5th) | 159 (36) | 85 (38.80) | 74 (33.20) | |
| Secondary (6th to 12th grade) | 26 (5.90) | 13 (5.90) | 13 (5.80) | |
| Higher (Upper than 12 grade) | 11 (2.50) | 8 (3.70) | 3 (1.30) | |
| **Personal income/month (US$)[b]** | 24 (21.33, 26.67) | 24 (21.45, 26.67) | 24 (20.41, 26.70) | 0.161[‡] |
| **Smoking [n, (%)]** | | | | |
| Yes | 91 (20.60) | 37 (16.90) | 54 (24.20) | 0.057 [†] |
| No | 351 (79.40) | 182 (83.10) | 169 (75.80) | |

[a, b] Results are presented as mean ± SD and median (25th percentile, and 75th percentile), respectively. Abbreviation: BMI, body mass index. BMI was calculated as body weight (Kg) divided by body height squared (m²).

[*]p, [†]p and [‡]p values were from the independent sample t-test, chi-square test, and Mann-Whitney U test, respectively. +Others included village doctor, security guard, banker, teacher, rickshaw puller, and tailor.

## Causal mediation analysis

Table 3 shows the results of the causal mediation analysis that investigated the periostin levels as a mediator of the association between arsenic exposure levels and skin lesions. Both the direct effect (DE), and indirect effect (IE, through the mediator periostin levels) of arsenic on skin lesions were significant. For example, a unit increase in log nail arsenic level was directly associated with a 9.87-fold increase in the odds of skin lesions [OR (95% CI): 9.87 (5.64, 17.46)], whereas the same increase in log nail arsenic indirectly contributed to another significant 73% increase in the odds of skin lesions [OR (95% CI): 1.73 (1.39, 2.29)] through periostin levels.

## Distribution of periostin levels across the stages of skin lesions

First, we examined the distribution of periostin levels between the groups of participants with and without skin lesions (Fig 3A). The median (IQR) periostin level was significantly higher ($p < 0.001$) for the group of participants with skin lesions compared to those without skin lesions [68.80 (44.96, 105.52) vs. 37 (23.44, 53.42)]. Further, we examined the distribution of periostin levels by the stages (early and advanced) of skin lesions (Fig 3B). The median (IQR) periostin levels among the participants with early-stage and advanced stage skin lesions were

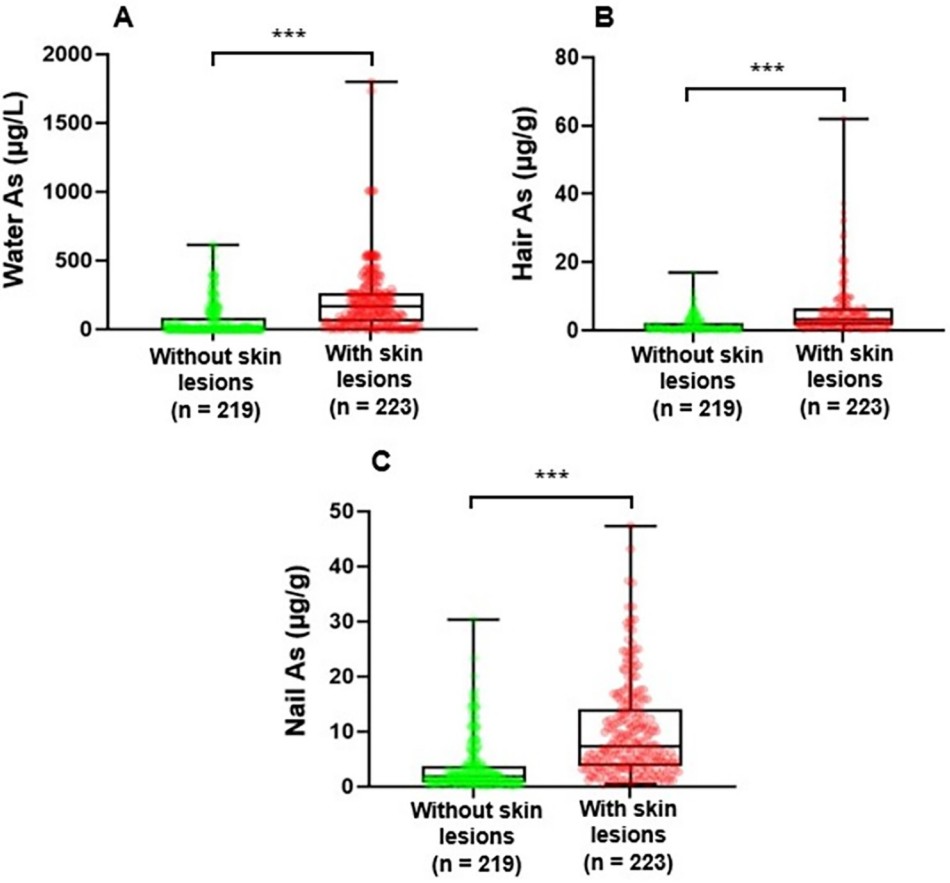

**Fig 2. Comparisons of arsenic exposure levels between the groups of participants with and without skin lesions using box and whisker plots.** Comparisons of water (A), hair (B), and nail (C) arsenic levels between the groups of participants with and without skin lesions. The center line indicates the median value, the box contains the 25th to 75th percentiles of the dataset. The whiskers mark the minimum and maximum values. The dots show the individual data points. Mean rank differences were assessed by using the Mann-Whitney U test between each skin lesions group. ***$p < 0.001$.

64.32 (41.68, 89.60) and 79.24 (49.72, 130.80), respectively. In high arsenic-exposure areas, many individuals were found to have subclinical effects (without visible skin lesions) [1, 62, 63]. In our study, it is likely that many participants without apparent skin lesions in the high-exposure areas were subclinical patients. Therefore, we further divided the participants

**Table 2. Association of arsenic exposure and serum periostin levels with the risk of skin lesions.**

| Variables | Skin lesions | |
|---|---|---|
| | OR (95% CI) | *p*-value |
| **Water As (µg/L)** | 3.41 (2.66, 4.37) | <0.001 |
| **Hair As (µg/g)** | 10.22 (6.41, 16.30) | <0.001 |
| **Nail As (µg/g)** | 15.66 (9, 27.24) | <0.001 |
| **Periostin (ng/mL)** | 1.03 (1.02, 1.04) | <0.001 |

Abbreviation: OR, Odds ratio. $Log_{10}$-transformed values of As concentrations were used. Adjusted for age, sex (male used as a referent group), BMI, smoking habit (non-smokers used as a referent group), and education (no formal education used as a referent group).

**Table 3. Direct and indirect effects of arsenic exposure levels (mediated by periostin levels) on skin lesions.**

| Variables | Skin lesions | | |
|---|---|---|---|
| | Direct effect | Indirect effect | *p*-value |
| Water As (μg/L) | 2.80 (2.18, 3.60) | 1.28 (1.17, 1.46) | <0.001 |
| Hair As (μg/g) | 7.24 (4.48, 11.70) | 1.63 (1.35, 2.12) | <0.001 |
| Nail As (μg/g) | 9.87 (5.64, 17.46) | 1.73 (1.39, 2.29) | <0.001 |

Odds ratios and corresponding 95% confidence intervals are presented for the direct, and indirect (mediated by periostin levels) effects, and are presented separately for each As exposure. $Log_{10}$-transformed values of As concentrations and periostin levels were used. Data were adjusted for age, sex (male used as a referent group), BMI, smoking habit (non-smokers used as a referent group), and education (no formal education used as a referent group).

without skin lesions into two groups: participants without skin lesions in the low-exposure area and participants without skin lesions in high-exposure areas and then examined the distribution of periostin among the four groups (S2 Fig). The median (IQR) periostin levels of the groups of participants without skin lesions in the low-exposure area, without skin lesions in high exposure areas, and with early-stage and advanced-stage skin lesions were 33.20 (20.24, 46.08), 46.78 (27.49, 73.04), 64.32 (41.68, 89.60) and 79.24 (49.72, 130.80), respectively.

## Associations of arsenic exposure and serum periostin levels with the risk of different stages of skin lesions

Table 4 shows the associations of arsenic exposure and serum periostin levels with stages of skin lesions. Each log unit increase in arsenic exposure level was gradually and significantly associated with the odds of early-stage and advanced-stage skin lesions, and each unit increase in periostin levels significantly increased the odds of both stages of skin lesion (OR = 1.02, $p <$ 0.001 for early-stage skin lesions, and OR = 1.03, $p <$ 0.001 for advanced-stage skin lesions).

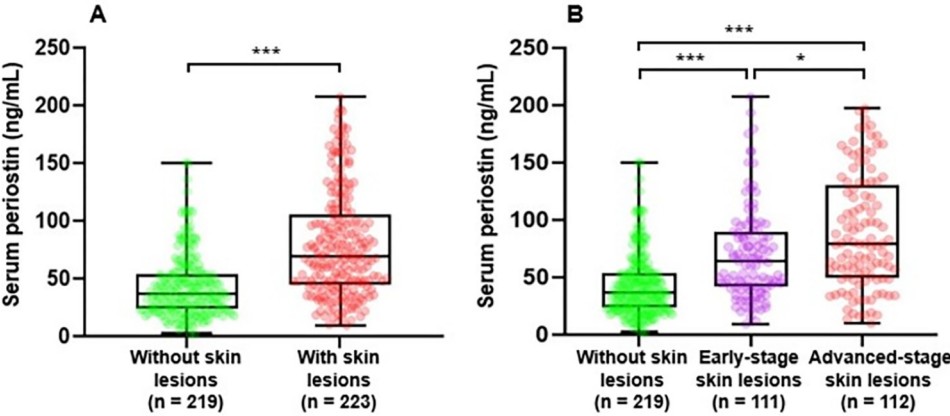

**Fig 3. Comparison of the distribution patterns of serum periostin in the different stages of skin lesions using box and whisker plots.** (A) Comparison between the groups of participants with and without skin lesions (B) Comparison among the participants without, early- and advanced-stage skin lesions groups. The center line indicates the median value, the box contains the 25th to 75th percentiles of the dataset. The whiskers mark the minimum and maximum values. The dots show the individual data points. Mean rank differences were assessed by using the Mann-Whitney U test (A), and the Kruskal-Wallis test followed by Dunn-Bonferroni post hoc test (B) between each skin lesions group. ***$p < 0.001$; *$p < 0.05$.

**Table 4. Association of arsenic exposure and periostin levels with the risk of different stages of skin lesions.**

| Variables | Skin lesions | | | |
|---|---|---|---|---|
| | Early-stage skin lesions | | Advanced-stage skin lesions | |
| | OR (95% CI) | *p*-value | OR (95% CI) | *p*-value |
| Water As (µg/L) | 2.93 (2.22, 3.88) | <0.001 | 4.12 (2.96, 5.74) | <0.001 |
| Hair As (µg/g) | 8.06 (4.80, 13.53) | <0.001 | 13.09 (7.48, 22.92) | <0.001 |
| Nail As (µg/g) | 13.10 (7.01, 24.48) | <0.001 | 18.65 (9.72, 35.78) | <0.001 |
| Periostin (ng/mL) | 1.02 (1.02, 1.03) | <0.001 | 1.03 (1.03, 1.04) | <0.001 |

Results were derived from multinomial logistic regression analysis with the three-category outcome (referent = without skin lesions). Abbreviation: OR, Odds ratio. Log$_{10}$-transformed values of As concentrations were used. Each row in this table was from separate models adjusted for age, sex (male used as a referent group), BMI, smoking habit (non-smoker used as a referent group), and education (no formal education used as a referent group).

## Comparisons of the levels of type 2 cytokines and IgE in the different stages of skin lesions

In our previous study [45], we showed the positive associations of arsenic-related type 2 cyto-kines (IL-4, IL-5, IL-13, and eotaxin), and IgE with periostin levels [45]. In this study, we examined whether type 2 cytokines and IgE levels were increased with the increasing severity of skin lesions. Therefore, we compared the medians (IQR) of IL-4, IL-5, IL-13, eotaxin, and IgE levels among the groups of participants without skin lesions, to those with early-stage and advanced-stage skin lesions (Fig 4 and S2 Table). Except for IL-4 in advanced-stage skin lesions, there were trends in increasing serum type 2 cytokines and IgE levels with the progression of the skin lesions. The median (IQR) of IL-4, IL-5, IL-13, eotaxin, and IgE levels were significantly higher in the early-and advanced-stage skin lesions compared to the group of participants without skin lesions.

## Discussion

Although periostin is involved in the pathogenesis of various types of skin diseases, the roles of periostin in arsenic-induced skin lesions have not been studied yet. This study is the first to show an association between serum periostin levels and the risk of arsenic-induced skin lesions in individuals who were chronically exposed to arsenic.

In our previous study [45], we showed a positive association between arsenic exposure and serum periostin levels. We used the same population and the same values of serum periostin and arsenic exposure levels for the present study. In this study, we found a dose-dependent association between arsenic exposure and the risk of skin lesions (Table 2). The associations between prolonged exposure markers, i.e., nails arsenic concentrations and the risk of skin lesions were in good agreement with the previous studies showing that prolonged exposure to moderate to high concentrations of arsenic are associated with skin lesions [64]. Elevated serum periostin levels were associated with the increased risk of arsenic-induced skin lesions (Table 2). Supporting these associations, causal mediation analysis also showed that the indirect effect of arsenic on skin lesions i.e., the effect mediated through periostin was statistically significant (Table 3). All these results suggest that periostin is implicated in arsenic-related skin lesions.

Previously, several studies have explained the roles of periostin in inflammatory skin diseases, particularly, atopic dermatitis (AD) and scleroderma [37, 39, 40]. Serum periostin level is correlated with the severity of AD and the highest levels were observed in AD patients with

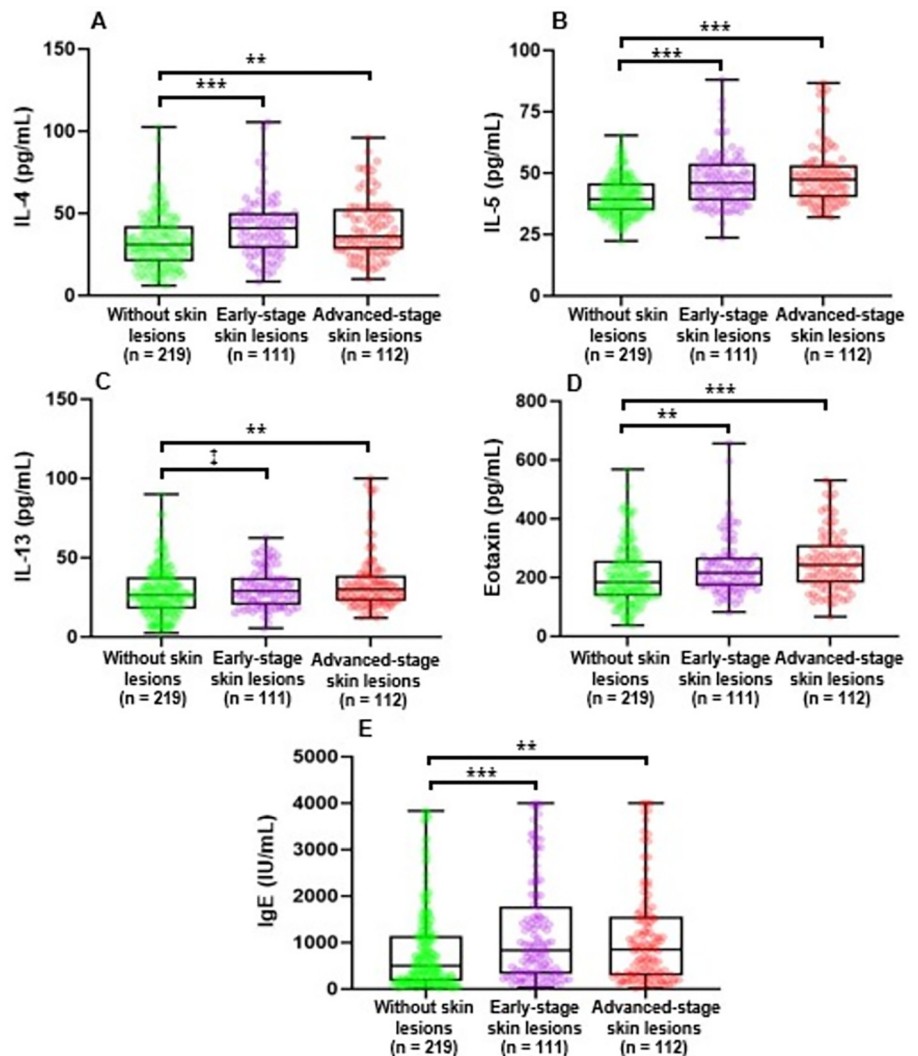

**Fig 4. Comparisons of the levels of type 2 cytokines and IgE in the different stages of skin lesions using box and whisker plots.** Comparisons of IL-4 (A), IL-5 (B), IL-13 (C), eotaxin (D), and IgE (E) levels between the groups of participants with and without skin lesions. The center line indicates the median value, the box contains the 25th to 75th percentiles of the dataset. The whiskers mark the minimum and maximum values. The dots show the individual data points. Mean rank differences were assessed by the Kruskal-Wallis test followed by Dunn-Bonferroni post hoc test between each skin lesions group. $^{***}p < 0.001$; $^{**}p < 0.01$; $^{‡}p = 0.075$.

skin lichenification [35]. An elevated periostin level in these inflammatory skin diseases is a result of dysregulated immune responses. Dysregulated immune responses cause the excessive production of type 2 cytokines, growth factors, and other inflammatory molecules implicated in hyperplasia and uncontrolled differentiation of the epidermis [40, 65]. The periostin produced by fibroblasts binds with integrin receptors on the keratinocytes. Receptor engagement by periostin ultimately activates keratinocytes and induces the production of thymic stromal lymphopoietin (TSLP) [65]. The TSLP then stimulates the production of type 2 cytokines along with periostin by activating and differentiating a specific subset of dendritic cells [28, 40, 65]. This amplification process of periostin further potentiates the activation and proliferation of keratinocytes and exacerbates inflammation in AD [65]. In agreement with the roles of periostin in AD, our results also showed that periostin levels were increased with the advancement

of the pathological condition in the skin, such as the levels were significantly higher in the group of participants with early-stage skin lesions compared to the group of participants without skin lesions which were further enhanced in the more advanced pathological condition (advanced-stage skin lesions) (Fig 3B). These results were supported by the risk analysis of early- and advanced-stage skin lesions with periostin which showed that risk of advanced-stage skin lesions was more strongly associated with the serum periostin levels than that of early-stage skin lesions (Table 4). The risk of skin lesions by periostin observed in our study was corresponded well with the other studies showing the association between periostin and other health outcomes [66–68].

We previously found that arsenic-related periostin levels were found linked to the levels of type 2 cytokines and serum IgE [45]. Interestingly, along with the elevated serum periostin levels with the severity of skin lesions, in this present study, we found the general trends in increasing serum type 2 cytokines (IL-4, IL-5, IL-13, and eotaxin) and IgE levels with the progression of skin lesions (Fig 4 and S2 Table). All these associations suggested that arsenic exposure-related increased periostin levels may result from the activation of type 2 immune response. Type 2 cytokines are mainly related to the allergic response and helminthic infection. We previously demonstrated an association between arsenic exposure and the features of allergic asthma [46]. However, we could not clarify how arsenic exposure is associated with allergic diseases. Based on the associations of arsenic exposure levels with several blood biomarkers of epithelial and endothelial dysfunction, such as VEGF and MMPs, we hypothesized that arsenic exposure increased the susceptibilities to environmental allergens through increasing vascular and epithelial permeability of the allergens [45, 46, 69, 70]. Interestingly, withdrawal of arsenic from drinking water reduces skin lesions [71, 72]. Based on this finding, it can be assumed that withdrawal of arsenic from drinking water may reduce the susceptibility to allergens by improving epithelial or vascular integrity that may cause the reduction of skin lesions.

Periostin might act on fibroblasts and other cells to promote arsenic-induced skin tissue remodeling as periostin has been reported to be involved in the activation and proliferation of fibroblast cells [33, 73]. Periostin also activates epithelial cells and stimulates the production of TGF-β which stimulates collagen production [74]. Acting on eosinophils, periostin causes transmigration, chemotaxis, and adhesion of eosinophils at the site of the inflammation [75]. As a matricellular protein, periostin causes collagen cross-linking through its interaction with other extracellular matrix proteins and enzymes leading to tissue remodeling [76]. Because of its pleiotropic properties, the role of periostin in arsenic-induced skin lesions may be complicated and diverse.

Periostin is involved in the initiation of the tumor microenvironment. High serum periostin levels have been observed in several types of cancers [25, 28, 77, 78]. Additionally, high levels of periostin in serum are associated with the poor prognosis of a variety of cancers [21, 27, 79–81]. Periostin is also involved in the invasion and metastasis of some cancers, including squamous cell carcinoma [31, 82]. Skin lesions, particularly hyperkeratotic skin lesions, may be precancerous and even the indicators for future development of arsenic-induced several cancers in internal organs [13–15]. Therefore, the association of the circulating periostin levels with advanced-stage skin lesion i.e., hyperkeratotic skin lesions observed in our study suggest the possible involvement of periostin in the pathogenesis of arsenic-induced skin cancer.

Subclinical effects (without visible skin lesions) may be more widespread than the clinical effects in high-arsenic exposure areas [1, 62, 63]. It is likely that many participants without apparent skin lesions in the high-exposure areas were subclinical patients. Epidemiological studies indicate that prolonged exposure to arsenic via drinking water at >100 μg/L induces skin lesions [52, 83]. Approximately 60% of our participants without skin lesions in the high-arsenic exposure areas had drinking water arsenic concentrations at >100 μg/L. All our participants had at least 5 year-residencies in their respective areas. Moreover, 76% of participants'

hair arsenic and 96% of participants' nail arsenic concentrations were greater than the toxic limit (1.00 μg/g). The average concentrations of hair and nail arsenic concentrations of these 76% and 96% of participants were 3.94 ± 2.69 μg/g and 6.75 ± 5.70 μg/g, respectively, which were much greater than those associated with subclinical effects [1, 84, 85]. Thus, it is assumed that prolonged exposure to such a high concentration of arsenic might initiate the pathogenesis of subclinical skin lesions that cannot be identified visually. Given this possibility, we further compared the medians of periostin levels in the four groups (participants without skin lesions in low-exposure areas, participants without skin lesions in high-exposure areas, and participants with early-stage and participants with advanced-stage skin lesions) (S2 Fig). Notably, the median periostin levels were significantly higher in the group of participants without skin lesions in the high-exposure areas compared to that of participants without skin lesions in the low-exposure area. Additionally, progressive and significant increases in periostin levels were observed with the increasing severity of diseases even if we don't consider the group of participants without skin lesions from the low-exposure area (S2 Fig). These results suggest that periostin may be involved in the pathogenesis from the preclinical to the advanced stages of skin lesions.

This study had several limitations. First, in this study, circulating periostin levels do not clarify the source of its expression. Direct histopathological examination of periostin in skin samples would be a more specific approach to reveal the roles of periostin in arsenic-induced skin lesions. Second, because of its cross-sectional nature and the limited number of measured covariates, this study could not totally explain the mediating effects of periostin in arsenic-induced skin lesions. There might be residual confounding due to unmeasured confounders. Prospective cohort studies are required in the future to explain the mediating effects of periostin in the pathogenesis of skin lesions. Third, we could not completely exclude the possibility of bias in identifying the subjects with skin lesions. In some cases, over diagnosis or under diagnosis could occur as it was difficult to differentiate arsenic-induced skin lesions from non-arsenic-related skin lesions. Arsenic-induced melanosis and hyperkeratosis could mimic other non-arsenical skin lesions. However, skin lesion cases were identified in the high-arsenic exposure areas. Approximately 70% of drinking water sources (tube well water) of our participants in high-exposure areas exceeded 50 μg/L of arsenic and more than 60% of participants had drinking water arsenic levels greater than 100 μg/L. Drinking water of only 10% of participants contained low levels (<10 μg/L) of arsenic. All advanced-stage patients had both hyperkeratosis and melanosis. When melanosis and keratosis appear together, they especially indicate arsenical toxicity [6]. We recruited participants who had been living in arsenic-endemic areas for at least five years. High prevalence of advanced-stage skin lesions, presence of high concentrations of arsenic in the drinking water, hair, and nail samples, and long-time residencies of the participants in arsenic-endemic areas thus suggested that there might be minimum bias in the diagnosis of arsenical skin lesions over non-arsenical skin lesions. Additionally, our results argue that if there was a substantial bias in the diagnosis, it would be unlikely to observe the progressive increase of the risk of skin lesions with increasing concentrations of arsenic and periostin levels (Tables 2 and 4). Fourth, this study has been conducted on a highly arsenic-exposed population who were from low socioeconomic conditions. Therefore, additional factors, such as nutritional levels, could affect the associations, which may limit the reproducibility of the results in other populations.

## Conclusion

This study showed a novel association between elevated levels of serum periostin and the risk of arsenic-induced skin lesions. Arsenic exposure-related periostin levels were progressively

increased with the severity of skin lesions. Additionally, there were general trends in increasing serum type 2 cytokines and IgE levels with the progression of the skin lesions. The results of this study suggest that periostin is involved in the pathogenesis and progression of arsenic-induced skin lesions through dysregulating the type 2-mediated immune response. The arsenic-induced precancerous skin lesion is not only an indicator of future cancer in skin but may also be a predictor of the various types of cancers in other internal organs. Thus, the association of periostin with skin lesions observed in this study warrants further epidemiological and experimental research to explore how periostin is involved in the pathogenesis of skin lesions.

## Supporting information

**S1 Fig. Directed acyclic graph (DAG) representing the causal association between arsenic exposure levels and skin lesions mediated by periostin.** A (exposure): arsenic exposure metrics; M (mediator): periostin; C (confounders not affected by the exposure): age, sex, BMI, smoking, education; and Y (outcome): skin lesions.
(TIF)

**S2 Fig. Comparison of the distribution patterns of serum periostin in the different stages of skin lesions using box and whisker plots.** Comparison among the without skin lesions group in the low-exposure area and without, early- and advanced-stage skin lesions groups in the high-exposure areas. × indicates the mean and the straight line connecting the × indicates the mean line. Mean differences were assessed by using the Kruskal-Wallis test followed by Dunn-Bonferroni post hoc test between each skin lesions group. [a, b, c] Significant difference from without skin lesions group in the low-exposure area, without skin lesions group in the high-exposure areas, and early-stage skin lesions group, respectively. $^{***}p < 0.001$; $^{**}p < 0.01$; $^{*}p < 0.05$.
(TIF)

**S1 Table. Comparisons of arsenic exposure levels between the groups of participants with and without skin lesions.**
(DOCX)

**S2 Table. Comparisons of the levels of type 2 cytokines and IgE in the different stages of skin lesions.**
(DOCX)

## Author Contributions

**Conceptualization:** Khaled Hossain.

**Data curation:** Nazmul Haque, Jahidul Islam, Shahnur Alam.

**Formal analysis:** Abu Eabrahim Siddique, Abdus S. Wahed.

**Funding acquisition:** Aaron Barchowsky, Seiichiro Himeno, Khaled Hossain.

**Investigation:** Moriom Khatun, Selim Reza Tony, Md. Khalequzzaman Sarker, Isabela Kabir.

**Methodology:** Khaled Hossain.

**Project administration:** Zahangir Alam Saud.

**Resources:** Daigo Sumi.

**Supervision:** Khaled Hossain.

**Validation:** Shakhawoat Hossain.

**Visualization:** Khaled Hossain.

**Writing – original draft:** Moriom Khatun, Abu Eabrahim Siddique.

**Writing – review & editing:** Abdus S. Wahed, Aaron Barchowsky, Seiichiro Himeno, Khaled Hossain.

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
