## [Decision Letter · Decision Letter 0]

2 Nov 2022

PONE-D-22-27210Association between serum periostin levels and the severity of arsenic-induced skin lesionsPLOS ONE

Dear Dr. Hossain,

Thank you for submitting your manuscript to PLOS ONE. After careful consideration, we feel that it has merit but does not fully meet PLOS ONE’s publication criteria as it currently stands. Therefore, we invite you to submit a revised version of the manuscript that addresses the points raised during the review process.

We look forward to receiving your revised manuscript.

Kind regards,

J. Christopher States

Academic Editor

PLOS ONE

Journal Requirements:

2. We note that Figure 1 in your submission contain copyrighted images. All PLOS content is published under the Creative Commons Attribution License (CC BY 4.0), which means that the manuscript, images, and Supporting Information files will be freely available online, and any third party is permitted to access, download, copy, distribute, and use these materials in any way, even commercially, with proper attribution. For more information, see our copyright guidelines: http://journals.plos.org/plosone/s/licenses-and-copyright.

Additional Editor Comments:

There are several suggestions made by the reviewers to improve the presentation of the data, as well as the clarity and transparency of the methodology. Incorporating these suggestions and answers to the reviewer queries is essential in a revised manuscript.

Reviewers' comments:

Reviewer's Responses to Questions

**Comments to the Author**

1. Is the manuscript technically sound, and do the data support the conclusions?

Reviewer #1: Yes

Reviewer #2: Yes

2. Has the statistical analysis been performed appropriately and rigorously? 

Reviewer #1: Yes

Reviewer #2: Yes

3. Have the authors made all data underlying the findings in their manuscript fully available?

Reviewer #1: Yes

Reviewer #2: No

4. Is the manuscript presented in an intelligible fashion and written in standard English?

Reviewer #1: Yes

Reviewer #2: Yes

5. Review Comments to the Author

Reviewer #1: The manuscript entitled, “Association between serum periostin levels and the severity of arsenic-induced skin lesions” (PONE-D-22-27210) investigated the correlation between serum periostin levels, serum levels of type 2 cytokines and skin lesion severity in a highly arsenic exposed population from Bangladesh.

The manuscript is well written, and the methods are described with clarity. The statistical analyses are described and justified in sufficient details as well as conducted with impeccable rigor. The conclusions are derived properly from the in-depth data analysis. Overall, this manuscript adds novel and critical information to the literature on effects of arsenic exposure.

There are a few minor points that need to be clarified and data presentation can be improved through the following modifications.

1. The authors should include information on the response rate for the sample collection. This information is missing from this article, as well as the earlier article that they cite in relation to sample collection.

2. What measures were adopted to quantify and eliminate recall bias of the study participants during the generation of questionnaire based demographic data?

3. This manuscript will benefit from better visualization of the data rather than tabulation. As such, I suggest that the authors move tables 2 and 6 to the supplementary data and replace them with box and whisker plots showing individual data points for all samples in the main manuscript for better clarity. Additionally, they should also update the box plot in Figure 2 to show all the individual data points in each group in addition to the box plots.

Reviewer #2: Major comments

1. The odds ratios reported for periostin seem very large. Is there prior research with regard to the association between periostin and other health outcomes that could demonstrate the plausibility of these values? Can some information be added to assist with interpretation, given that the periostin values were log-transformed?

2. Given that the mediation analysis is one of the highlighted findings in this manuscript, the methods used for the mediation analysis need to be described in more detail. There are several different methods use to perform mediation analysis, including more traditional methods (which are not recommended) as well as more advanced, sophisticated methods. At least a discussion of the possibility of residual confounding between the exposure and mediator and the mediator and outcome should be mentioned. A formal mediation analysis also requires a prospective study ideally, and this limitation should also be mentioned in the context of the mediation analysis.

Minor comments

1. Why not add periostin to Table 2?

2. It is not described, how the covariates were modeled/controlled for. For example, was education treated in categories or continuously? It is commonplace to add this information to table footnotes.

3. Please use a consistent number of decimal places in text and tables. Even if the final decimal place is a zero, it should be included to show the degree of precision.

Typos

1. Table four, odds ratio should be “odds ratios”

2. Bottom of p. 10 (line 218), n=112 represents ¼ of the total participants, not ¼ of the participants with skin lesions.

3. A 2.3-fold increase (line 248) does not correspond to an odds ratio of 9.87.

4. Line 265, “lesions in low-exposure area” should be “lesions in the low-exposure area.

5. Line 281, “…without skin lesions, early-stage and advanced-stage…” should be “…without skin lesions, to those with early-stage and advanced-stage…”

6. No comma is needed after “approximately” in line 389.

6. PLOS authors have the option to publish the peer review history of their article (what does this mean?). If published, this will include your full peer review and any attached files.

Reviewer #1: **Yes: **Mayukh Banerjee

Reviewer #2: No

---

## [Author Response · Author response to Decision Letter 0]

4 Dec 2022

Responses to the reviewers’ comments

Reviewer # 1

1. The authors should include information on the response rate for the sample collection. This information is missing from this article, as well as the earlier article that they cite in relation to sample collection.

Response: For this study and our two previous studies [Tony et al., 2022; Rahman et al., 2021], we could not calculate the response rate as we did not recruit any new individuals. Study participants recruitment was done in our previous study [Siddique et al., 2020] which showed the association between arsenic exposure and characteristic features of asthma. The overall response rate of that study [Siddique et al., 2020] was 92% (93% from arsenic-endemic areas, and 91% from non-endemic area). We have included this information in the “Materials and methods” section (please see lines: 127-130) of the revised manuscript.

2. What measures were adopted to quantify and eliminate recall bias of the study participants during the generation of questionnaire based demographic data?

Response: The demographic information collected through questionnaires were age, sex, occupation, education, personal income, and smoking and used them as variables in our study. These variables are generally known not to be affected by recall bias.

3. This manuscript will benefit from better visualization of the data rather than tabulation. As such, I suggest that the authors move tables 2 and 6 to the supplementary data and replace them with box and whisker plots showing individual data points for all samples in the main manuscript for better clarity. Additionally, they should also update the box plot in Figure 2 to show all the individual data points in each group in addition to the box plots.

Response: As per reviewer’s suggestion, we have included Tables 2 and 6 in the supplement and included their graphical representation in the main manuscript (please see Fig 2 and Fig 4 in the revised manuscript). Additionally, individual data points have been included in Fig 3 (Fig 2 in the previous manuscript).

Reviewer # 2

Major comments

1. The odds ratios reported for periostin seem very large. Is there prior research with regard to the association between periostin and other health outcomes that could demonstrate the plausibility of these values? Can some information be added to assist with interpretation, given that the periostin values were log-transformed?

Response: We used log transformed periostin value in our regression model to examine the association between periostin levels and the risk of skin lesion (Table 3 and Table 5 in the previous manuscript) to improve the estimation quality. If we don’t use log transformed values of periostin, the odds ratios will be small but corresponded well with the other studies showing the association between periostin and other health outcomes [Elhady et al., 2017; Yavuz et al., 2021; Ji et al., 2017]. We have included one statement in the discussion section regarding the similarities of our results with other studies on different health outcomes (please see lines: 334-335). Since periostin is a covariate here, normality assumption is not needed. We have revised Table 3 and Table 5 (Table 2 and Table 4 in the revised manuscript) with original (without log transformed) values of periostin.

2. Given that the mediation analysis is one of the highlighted findings in this manuscript, the methods used for the mediation analysis need to be described in more detail. There are several different methods used to perform mediation analysis, including more traditional methods (which are not recommended) as well as more advanced, sophisticated methods. At least a discussion of the possibility of residual confounding between the exposure and mediator and the mediator and outcome should be mentioned. A formal mediation analysis also requires a prospective study ideally, and this limitation should also be mentioned in the context of the mediation analysis.

Response: Thank you for the valuable suggestion. We represented the causal relationship using direct, and indirect (mediated through periostin) effects which were estimated using the standard Baron-Kenny regression-based approach [Valeri and Vanderweele, 2013; VanderWeele and Vansteelandt, 2014]. We have postulated a directed acyclic graph (DAG) which has been included in the revised Manuscript (Fig S2). We have provided a short description on the mediation analysis in the “Statistical analysis” section (please see lines: 197-210) of the revised manuscript. We have also included the limitation of the analysis in the discussion section (please see lines: 396-400). 

Minor comments

1. Why not add periostin to Table 2?

Response: The reviewer #1 has suggested that we move Table 2 into supplementary (Table S1 in the revised manuscript) and replace them with box and whisker plots. If we add periostin data to Table S1 (Table 2 in the previous manuscript) and present them with box and whisker plots, it will be redundant to another Figure (Fig 3A in the revised manuscript). To avoid the redundancy, the periostin data have not been presented in Table S1 and Fig 2.

2. It is not described, how the covariates were modeled/controlled for. For example, was education treated in categories or continuously? It is common place to add this information to table footnotes.

Response: We described how the covariates were modeled in the statistical section (please see lines: 189-192 in the revised manuscript). We used education as categorical variable which was mentioned in the statistical section (please see lines: 177-179 in the revised manuscript). We also added this information to the table footnotes (Table 2, 3, and 4 in the revised manuscript).

3. Please use a consistent number of decimal places in text and tables. Even if the final decimal place is a zero, it should be included to show the degree of precision.

Response: We have revised the text and tables according to the suggestion of the reviewer.

Typos

1. Table four, odds ratio should be “odds ratios”

Response: We have fixed the error (Table 3 in the revised manuscript).

2. Bottom of p. 10 (line 218), n=112 represents ¼ of the total participants, not ¼ of the participants with skin lesions.

Response: We have revised it (please see lines: 230-231 in the revised manuscript).

3. A 2.3-fold increase (line 248) does not correspond to an odds ratio of 9.87.

Response: We have revised it (please see line: 259 in the revised manuscript).

4. Line 265, “lesions in low-exposure area” should be “lesions in the low-exposure area.

Response: We have revised it (please see line: 277 in the revised manuscript).

5. Line 281, “…without skin lesions, early-stage and advanced-stage…” should be “…without skin lesions, to those with early-stage and advanced-stage…”

Response: We have revised it (please see line: 292 in the revised manuscript).

6. No comma is needed after “approximately” in line 389.

Response: We have removed the comma (Please see line: 405 in the revised manuscript).

---

## [Decision Letter · Decision Letter 1]

19 Dec 2022

Association between serum periostin levels and the severity of arsenic-induced skin lesions

PONE-D-22-27210R1

Dear Dr. Hossain,

We’re pleased to inform you that your manuscript has been judged scientifically suitable for publication and will be formally accepted for publication once it meets all outstanding technical requirements.

Kind regards,

J. Christopher States

Academic Editor

PLOS ONE

Additional Editor Comments (optional):

Reviewers' comments:

Reviewer's Responses to Questions

**Comments to the Author**

1. If the authors have adequately addressed your comments raised in a previous round of review and you feel that this manuscript is now acceptable for publication, you may indicate that here to bypass the “Comments to the Author” section, enter your conflict of interest statement in the “Confidential to Editor” section, and submit your "Accept" recommendation.

Reviewer #1: All comments have been addressed

Reviewer #2: All comments have been addressed

2. Is the manuscript technically sound, and do the data support the conclusions?

Reviewer #1: Yes

Reviewer #2: Yes

3. Has the statistical analysis been performed appropriately and rigorously? 

Reviewer #1: Yes

Reviewer #2: Yes

4. Have the authors made all data underlying the findings in their manuscript fully available?

Reviewer #1: Yes

Reviewer #2: Yes

5. Is the manuscript presented in an intelligible fashion and written in standard English?

Reviewer #1: No

Reviewer #2: Yes

6. Review Comments to the Author

Reviewer #1: (No Response)

Reviewer #2: The web address given for data availability under Figshare led to a dataset from an unrelated manuscript. Please correct the address or provide more detailed instructions for how to access and search Figshare from a typical web browser.

7. PLOS authors have the option to publish the peer review history of their article (what does this mean?). If published, this will include your full peer review and any attached files.

Reviewer #1: **Yes: **Mayukh Banerjee

Reviewer #2: No

---

## [Editor Report · Acceptance letter]

26 Dec 2022

PONE-D-22-27210R1 

Association between serum periostin levels and the severity of arsenic-induced skin lesions 

Dear Dr. Hossain:

I'm pleased to inform you that your manuscript has been deemed suitable for publication in PLOS ONE. Congratulations! Your manuscript is now with our production department. 

Kind regards, 

on behalf of

Dr J. Christopher States 

Academic Editor

PLOS ONE